# Fast surface reconstruction algorithm with adaptive step size

**Jingguo Dai[1], Yeqing Yi[2]\*, Chengzhi Liu[3]\***

**1** School of Artificial Intelligence, Guangzhou Huashang College, Guangzhou, China, **2** School of Information Engineering, Shaoguan University, Shaoguan, China, **3** School of Mathematics and Finance, Hunan University of Humanities, Science and Technology, Loudi, China

\* yeqingyi@126.com (YY); it-rocket@163.com (CL)

**Data Availability Statement:** All relevant data are within the article.

**Funding:** The author(s) received no specific funding for this work.

## Abstract

In (Dai et al. 2023), the authors proposed a fast algorithm for surface reconstruction that converges rapidly from point cloud data by alternating Anderson extrapolation with implicit progressive iterative approximation (I-PIA). This algorithm employs a fixed step size during iterations to enhance convergence. To further improve the computational efficiency, an adaptive step size adjustment strategy for surface reconstruction algorithm is investigated. During each iteration, the step size is adaptively chosen based on the current residual—larger residuals may necessitate larger steps, while smaller ones might permit smaller steps. Numerical experiments indicate that, for equivalent reconstruction errors, the adaptive step size algorithm demands substantially fewer iterations and less computation time than the fixed step size approach. These improvements robustly enhance computational performance in surface reconstruction, offering valuable insights for further research and applications.

## 1 Introduction

Surface reconstruction is the process of creating geometric models and capturing appearance information for three-dimensional scenes from two-dimensional images, point cloud data, or pixel data [1, 2]. This technique is widely applied in fields such as computer vision, computer graphics, virtual reality, and cultural heritage preservation. The primary goal of surface reconstruction is to convert discrete point clouds or pixel data into continuous three-dimensional surface models. The aim is to produce smooth and accurate reconstructions that closely mimic the actual surfaces of the objects.

The scanning technology is advancing rapidly, and with the continuous development of scanning techniques, point cloud data is experiencing rapid expansion. The large-scale point cloud datasets present a series of challenges for reconstruction algorithms. On the one hand, the vast amount of point cloud data requires efficient, accurate, and noise-resistant algorithms for processing and analysis. On the other hand, as the scale of point cloud data increases, modeling complex scenes becomes more challenging, requiring the overcoming of additional obstacles. In recent years, researchers have been continuously exploring and summarizing,

**Competing interests:** The authors have declared that no competing interests exist.

achieving significant progress. These advancements include methods such as constructing triangle meshes for surface reconstruction based on triangulation [3, 4], pixel-based approaches utilizing image information for surface reconstruction [5], implicit function-based methods representing surfaces [6–9], surface fitting methods that generate smooth surface models through fitting algorithms [1, 10], and machine learning approaches [2, 11] that learn the patterns and features of surface reconstruction using training data and models.

Implicit progressive iterative approximation (I-PIA), as detailed in [12], is an iterative method employed for the reconstruction of surfaces. Its primary objective is to recover a continuous surface representation from discrete point cloud data or other surface sampling data through iterative approximation. The process is as follows: First, an initial estimation of an implicit function is constructed from the discrete point cloud data or other surface sampling data. Then, a series of iterative steps are performed to gradually improve the quality of the implicit function approximation, aiming to more accurately describe the shape and attributes of the surface. The update rules during the iteration are typically based on local or global data fitting or optimization criteria. These criteria can utilize geometric features, normal information, curvature properties, etc., of the point cloud data to optimize the parameters or shape of the implicit function. The iteration process continues until a predefined convergence criterion is met or a certain number of iterations is reached. The I-PIA excels at managing complex geometric forms and noisy datasets. It has the ability to extract smooth, continuous surface representations even from sparse point cloud data, demonstrating flexibility in dealing with surface irregularities such as holes and boundaries. Moreover, I-PIA can adjust to data of varying densities and distributions, effectively managing non-uniform sampling situations. These capabilities have made I-PIA a preferred technique in the realm of surface reconstruction.

However, I-PIA also encounters several challenges. Its computational complexity is typically high, particularly when processing large-scale point cloud data. Since an iterative optimization process is necessary, each iteration involves calculating the distance from points to the surface, assessing the reconstruction error, and updating parameter estimates. For large datasets, this can lead to prolonged computation times and substantial storage requirements. To address the limitations of I-PIA, researchers have proposed various techniques, including relaxed implicit stochastic algebraic reconstruction [13], stochastic I-PIA [14], accelerated I-PIA based on the conjugate gradient method [15], and I-PIA with acceleration factors [16]. These techniques aim to enhance the efficiency and scalability of I-PIA by introducing relaxation strategies, stochastic approaches, acceleration factors, or leveraging the conjugate gradient method.

In the field of surface reconstruction, an accelerated algorithm was introduced in [17] that alternates between the Anderson extrapolation method [18, 19] and I-PIA, swiftly converging to the implicit zero-value surface of point cloud data. This iterative approach incorporates a step size parameter designed to expedite convergence. While the method utilizes a fixed step size adjustment strategy and includes a mechanism for selecting the optimal fixed step size, leading to satisfactory convergence outcomes, it is not without its limitations. To enhance the convergence performance of iterative methods, two primary strategies are employed: fixed step size and asynchronous step size [16, 17, 20–25]. The fixed step size strategy, despite its simplicity and ease of implementation, often falls short in delivering substantial improvements. Conversely, asynchronous strategies offer the flexibility to adjust the step size in response to the specific conditions of each iteration. This adaptability results in superior robustness and faster convergence rates [21]. Among asynchronous strategies, the adaptive step size stands out by dynamically adjusting the step size based on the residual of the current iteration [20, 22, 23, 25]. This dynamic adjustment offers significant computational efficiency benefits. By continuously monitoring the algorithm's progress in real-time and making

corresponding adjustments to the step size, the adaptive step size strategy facilitates a more rapid approach to the optimal solution. Building upon the foundation laid in [17], this paper introduces an adaptive step size adjustment strategy. The aim is to further improve the convergence speed of the surface reconstruction algorithm, thereby enhancing its computational efficiency. This strategy dynamically adjusts the step size for each iteration based on the progress of the current iteration, which can lead to more effective and faster convergence to the desired solution.

The structure of this paper is as follows: Section 2 introduces the progressive iterative approximation method for implicit B-spline surface reconstruction. Section 3 presents the asynchronous implicit progressive iterative approximation method and the asynchronous Anderson extrapolation method, detailing how these two methods alternate to achieve a fast algorithm for implicit surface reconstruction. For dealing with noisy point cloud data, Section 4 introduces a regularized adaptive implicit surface reconstruction algorithm. Section 5 showcases the numerical experimental results. The final section provides a summary of the paper.

## 2 Preliminary

### 2.1 Implicit B-spline surface reconstruction

Consider the task of fitting a point cloud dataset $\{\mathbf{p}_r = (x_r, y_r, z_r), r = 1, 2, \cdots, n\}$ using an implicit B-spline function

$$f(\mathbf{p}) = f(x, y, z) = \sum_{i=1}^{N_x}\sum_{j=1}^{N_y}\sum_{k=1}^{N_z} c_{ijk}B_i(x)B_j(y)B_k(z) = 0, \qquad (1)$$

where $B_i(x)$, $B_j(y)$, and $B_k(z)$ are cubic B-spline basis functions, and $\{c_{ijk}\}$ for $i = 1, \cdots, N_x$, $j = 1, \cdots, N_y$, and $k = 1, \cdots, N_z$ are the control coefficients to be determined.

Typically, these coefficients $c_{ijk}$ can be derived by solving the system of equations

$$f(\mathbf{p}_r) = f(x_r, y_r, z_r) = 0, \quad r = 1, 2, \cdots, n,$$

or alternatively, by minimizing the sum of the squared algebraic distances through the optimization problem

$$\min_{c_{ijk}} \sum_{r=1}^{n} f^2(x_r, y_r, z_r).$$

To improve the robustness of surface reconstruction algorithms and to prevent trivial solutions, it is essential to introduce minor perturbations to the original dataset $\{\mathbf{p}_r, r = 1, 2, \cdots, n\}$. Specifically, we shift the original data points along the unit normal vectors $\{\mathbf{n}_r, r = 1, 2, \cdots, n\}$ in both directions by a small distance $\sigma$. This process yields an additional set of $2n$ data points

$$\mathbf{p}_l = \begin{cases} \mathbf{p}_r + \sigma\mathbf{n}_r, & l = n+r, \quad r = 1, 2, \cdots, n, \\ \mathbf{p}_r - \sigma\mathbf{n}_r, & l = 2n+r, \quad r = 1, 2, \cdots, n. \end{cases} \qquad (2)$$

The additional $2n$ data points are then combined with the original dataset to form an extended point cloud dataset $\{\mathbf{p}_l, l = 1, 2, \cdots, 3n\}$. Subsequently, the B-spline function (1) is adjusted to satisfy

$$f(\mathbf{p}_i) = \begin{cases} 0, & l = 1, 2, \cdots, n, \\ \varepsilon, & l = n+1, n+2, \cdots, 2n, \\ -\varepsilon, & l = 2n+1, 2n+2, \cdots, 3n. \end{cases} \qquad (3)$$

Here, $\varepsilon$ and $-\varepsilon$ denote the expected values of $f(\mathbf{p})$ for the inner and outer perturbed data points, respectively. This approach ensures that the surface reconstruction is both robust and accurate.

For convenience, let's define $B_{ijk}(x, y, z) = B_i(x)B_j(y)B_k(z)$ for $i = 1, 2, \cdots, N_x, j = 1, 2, \cdots, N_y$, and $k = 1, 2, \cdots, N_z$. By arranging the B-spline bases and control coefficients in lexicographical order, we can form two $N_xN_yN_z$-dimensional vectors

$$[B_{111}(x, y, z), B_{112}(x, y, z), \cdots, B_{N_xN_yN_z}(x, y, z)]$$

and

$$\mathbf{C} = [C_{111}, C_{112}, \cdots, C_{N_xN_yN_z}]^{\mathrm{T}}.$$

Consequently, the fitting problem (3) can be reformulated as a linear system of equations

$$\mathbf{BC} = \mathbf{b}, \tag{4}$$

where the right-hand vector $\mathbf{b}$ is

$$\mathbf{b} = [\underbrace{0, \cdots, 0}_{n}, \underbrace{\varepsilon, \cdots, \varepsilon}_{n}, \underbrace{-\varepsilon, \cdots, -\varepsilon}_{n}]^{\mathrm{T}},$$

and the coefficient matrix $\mathbf{B}$ is

$$\mathbf{B} = \begin{bmatrix} B_{111}(x_1, y_1, z_1) & B_{112}(x_1, y_1, z_1) & \cdots & B_{N_xN_yN_z}(x_1, y_1, z_1) \\ B_{111}(x_2, y_2, z_2) & B_{112}(x_2, y_2, z_2) & \cdots & B_{N_xN_yN_z}(x_2, y_2, z_2) \\ \vdots & \vdots & \ddots & \vdots \\ B_{111}(x_{3n}, y_{3n}, z_{3n}) & B_{112}(x_{3n}, y_{3n}, z_{3n}) & \cdots & B_{N_xN_yN_z}(x_{3n}, y_{3n}, z_{3n}) \end{bmatrix}.$$

The least squares solution to (4) provides the control coefficients necessary for fitting the surface. According to the theory of numerical algebra, $\mathbf{C}^*$ is the least squares solution to (4) if and only if

$$\mathbf{B}^{\mathrm{T}}\mathbf{BC}^* = \mathbf{B}^{\mathrm{T}}\mathbf{b}.$$

Moreover, the least squares solution can be expressed as

$$\mathbf{C}^* = \mathbf{B}^{\dagger}\mathbf{b},$$

where $\mathbf{B}^{\dagger}$ denotes the Moore-Penrose inverse of $\mathbf{B}$.

## 3 Adaptive implicit surface reconstruction algorithm

### 3.1 Asynchronous I-PIA

First, we initialize all the control coefficients of the implicit B-spline function to zero, yielding an initial implicit B-spline function

$$f^{(0)}(x, y, z) = \sum_{i=1}^{N_x} \sum_{j=1}^{N_y} \sum_{k=1}^{N_z} C_{ijk}^{(0)} B_i(x)B_j(y)B_k(z) = 0,$$

where $C_{ijk}^{(0)} = 0$ for $i = 1, 2, \cdots, N_x, j = 1, 2, \cdots, N_y$, and $k = 1, 2, \cdots, N_z$.

Next, we calculate the error

$$\delta_l^{(0)} = \begin{cases} 0 - f^{(0)}(x_l, y_l, z_l), & l = 1, 2, \cdots, n, \\ \varepsilon - f^{(0)}(x_l, y_l, z_l), & l = n+1, n+2, \cdots, 2n, \\ -\varepsilon - f^{(0)}(x_l, y_l, z_l), & l = 2n+1, 2n+2, \cdots, 3n \end{cases}$$

and the adjustment values for the control coefficients

$$r_{ijk}^{(0)} = \sum_{l=1}^{3n} B_i(x_l) B_j(y_l) B_k(z_l) \delta_l^{(0)}.$$

Therefore, the control coefficients of the implicit B-spline function can be updated as

$$C_{ijk}^{(1)} = C_{ijk}^{(0)} + \mu_0 r_{ijk}^{(0)}.$$

where $\mu_0$ is a non-negative real number, also known as the adjustment step size, introduced to ensure convergence. After obtaining the new control coefficients, a new implicit B-spline function can be generated, i.e.,

$$f^{(1)}(x, y, z) = \sum_{i=1}^{N_x} \sum_{j=1}^{N_y} \sum_{k=1}^{N_z} C_{ijk}^{(1)} B_i(x) B_j(y) B_k(z) = 0.$$

Assume that $f^{(m)}(x, y, z) = 0$ is the B-spline implicit function obtained after $m$ (where $m = 0$, $1, 2, \cdots$) iterations. Then, we can calculate the error

$$\delta_l^{(m)} = \begin{cases} 0 - f^{(m)}(x_l, y_l, z_l), & l = 1, 2, \cdots, n, \\ \varepsilon - f^{(m)}(x_l, y_l, z_l), & l = n+1, n+2, \cdots, 2n, \\ -\varepsilon - f^{(m)}(x_l, y_l, z_l), & l = 2n+1, 2n+2, \cdots, 3n, \end{cases} \tag{5}$$

and the adjustment values for the control coefficients

$$r_{ijk}^{(m)} = \sum_{l=1}^{3n} B_i(x_l) B_j(y_l) B_k(z_l) \delta_l^{(m)}.$$

Therefore, we can update the control coefficients

$$C_{ijk}^{(m+1)} = C_{ijk}^{(m)} + \mu_m r_{ijk}^{(m)}, \tag{6}$$

where the step size $\mu_m$ is a non-negative real number that varies with the iteration step $m$. So, the implicit B-spline function for the $(m + 1)$-th iteration can be generated, i.e.,

$$f^{(m+1)}(x, y, z) = \sum_{i=1}^{N_x} \sum_{j=1}^{N_y} \sum_{k=1}^{N_z} C_{ijk}^{(m+1)} B_i(x) B_j(y) B_k(z) = 0.$$

Consequently, we obtain a sequence of B-spline implicit functions $f^{(m)}(x, y, z) = 0$, $m = 0, 1, 2, \cdots$, which approximates the point set $\{\mathbf{p}_r\}_{r=1}^n$. During the iteration process, the step size varies with the iteration step $m$, hence this method is referred to as asynchronous implicit progressive iterative approximation (abbreviated as asynchronous I-PIA).

**Remark 1** *The I-PIA algorithm presented in* [12] *utilizes a fixed step size strategy, necessitating the computation of both the maximum and minimum eigenvalues of a matrix to determine the optimal step size. This process incurs substantial computational overhead. To mitigate this,*

the authors in [12] *suggest a more practical approach for selecting the step size, which, while reducing computational complexity, does not ensure a comparably swift convergence rate. This underscores the ongoing challenge of reconciling computational efficiency with the speed of convergence in real-world applications, offering valuable insights for enhancing algorithmic performance.*

**Remark 2**. *Unlike the I-PIA algorithm presented in* [12], *the step size $\mu_m$ used in this paper for updating the control coefficients is a real number that varies with the iteration step m. In other words, the value of $\mu_m$ differs at each step of the iterative process. This variability enables the proposed method to dynamically adjust the iteration step size based on the progress of the current iteration, thereby aiming to enhance the algorithm's robustness and convergence speed. It is evident that when $\mu_m$ (for $m = 0, 1, 2, \cdots$) in* Eq (6) *is fixed, the asynchronous I-PIA reverts to the I-PIA method described in* [1]. *The authors in* [17] *offer both theoretically optimal and practical values for fixed step sizes. In the following sections, we will present a step-size selection strategy tailored for the asynchronous I-PIA.*

Arranging the control coefficients $C_{ijk}^{(m)}$ in lexicographical order, we obtain an $N_x N_y N_z$-dimensional vector

$$\mathbf{C}^{(m)} = [C_{111}^{(m)}, C_{112}^{(m)}, \cdots, C_{11N_z}^{(m)}, \cdots, C_{N_x N_y N_z}^{(m)}]^{\mathrm{T}}.$$

The iterative process of updating the control coefficients can be expressed in matrix form as

$$\mathbf{C}^{(m+1)} = \mathbf{C}^{(m)} + \mu_m \mathbf{r}^{(m)} = (\mathbf{I} - \mu_m \mathbf{B}^{\mathrm{T}}\mathbf{B})\mathbf{C}^{(m)} + \mu_m \mathbf{B}^{\mathrm{T}}\mathbf{b}, \tag{7}$$

where **I** is the identity matrix of dimension $N_x N_y N_z$, and $\mathbf{r}^{(m)} = [r_{111}^{(m)}, r_{112}^{(m)}, \cdots, r_{11N_z}^{(m)}, \cdots, r_{N_x N_y N_z}^{(m)}]^{\mathrm{T}} = \mathbf{B}^{\mathrm{T}}(\mathbf{b} - \mathbf{B}\mathbf{C}^{(m)})$ is the residual.

It is worth noting that the iterative process in Eq (7) is equivalent to the Richardson Iteration method, enhanced with a dynamically adjusted weight $\mu_m$, as discussed in [20]. This method is traditionally used to solve the system of equations $\mathbf{B}^{\mathrm{T}}\mathbf{B}\mathbf{C} = \mathbf{B}^{\mathrm{T}}\mathbf{b}$. According to [20], various strategies exist for selecting the step size $\mu_m$, including the steepest descent method, the minimal residual method, among others. In this paper, we adaptively determine the weight by minimizing the residual, which allows for a more efficient and effective convergence.

In the $(m + 1)$-th iteration, the appropriate $\mu_m$ can be chosen by minimizing the squared $l^2$-norm of the residual, i.e.,

$$\mu_m = \arg \min_{\mu_m \in \mathbb{R}} \left\| \mathbf{B}^{\mathrm{T}}(\mathbf{b} - \mathbf{B}\mathbf{C}^{(m+1)}) \right\|_2^2. \tag{8}$$

Substituting (7) into (8), we obtain

$$\mu_m = \arg \min_{\mu_m \in \mathbb{R}} \left\| (\mathbf{I} - \mu_m \mathbf{B}^{\mathrm{T}}\mathbf{B})\mathbf{r}^{(m)} \right\|_2.$$

Let $g(\mu) = \left\| (\mathbf{I} - \mu\mathbf{B}^{\mathrm{T}}\mathbf{B})\mathbf{r}^{(m)} \right\|_2^2$. Then, it can be further expressed as

$$\begin{aligned} g(\mu) &= \left\| (\mathbf{I} - \mu\mathbf{B}^{\mathrm{T}}\mathbf{B})\mathbf{r}^{(m)} \right\|_2^2 \\ &= ((\mathbf{I} - \mu\mathbf{B}^{\mathrm{T}}\mathbf{B})\mathbf{r}^{(m)})^{\mathrm{T}}((\mathbf{I} - \mu\mathbf{B}^{\mathrm{T}}\mathbf{B})\mathbf{r}^{(m)}) \\ &= \left\| \mathbf{r}^{(m)} \right\|_2^2 - 2\mu(\mathbf{r}^{(m)})^{\mathrm{T}}(\mathbf{B}^{\mathrm{T}}\mathbf{B}\mathbf{r}^{(m)}) + \mu^2 \left\| \mathbf{B}^{\mathrm{T}}\mathbf{B}\mathbf{r}^{(m)} \right\|_2^2. \end{aligned}$$

Setting $\frac{dg}{d\mu} = 0$, the optimal step size $\mu_m$ for the $(m+1)$-th iteration of the asynchronous I-PIA method can be obtained as

$$\mu_m = \frac{(\mathbf{r}^{(m)})^{\mathrm{T}}(\mathbf{B}^{\mathrm{T}}\mathbf{B}\mathbf{r}^{(m)})}{\left\|\mathbf{B}^{\mathrm{T}}\mathbf{B}\mathbf{r}^{(m)}\right\|_2^2}.$$

## 3.2 The Anderson extrapolation for asynchronous I-PIA

Consider a weighted combination of $p + 1$ consecutive iterative sequences obtained by the asynchronous I-PIA, i.e.,

$$\bar{\mathbf{C}}^{(m)} = \mathbf{C}^{(m)} - \sum_{j=1}^{p} \omega_j^{(m)}(\mathbf{C}^{(m-p+j)} - \mathbf{C}^{(m-p+j-1)}), \tag{9}$$

where $\omega_j^{(m)}, j = 1, 2, \cdots, p$ are undetermined combination coefficients.

Define the combination coefficient vector

$$\mathbf{\Omega}^{(m)} = [\omega_1^{(m)}, \omega_2^{(m)}, \cdots, \omega_p^{(m)}]^{\mathrm{T}} \in \mathbb{R}^p,$$

the difference matrix of control coefficients

$$\mathbf{C}_m = [\mathbf{C}^{(m-p+1)} - \mathbf{C}^{(m-p)}, \mathbf{C}^{(m-p+2)} - \mathbf{C}^{(m-p+1)}, \cdots, \mathbf{C}^{(m)} - \mathbf{C}^{(m-1)}] \in \mathbb{R}^{N_x N_y N_z \times p},$$

and the difference matrix of residuals

$$\mathbf{R}_m = [\mathbf{r}^{(m-p+1)} - \mathbf{r}^{(m-p)}, \mathbf{r}^{(m-p+2)} - \mathbf{r}^{(m-p+1)}, \cdots, \mathbf{r}^{(m)} - \mathbf{r}^{(m-1)}] \in \mathbb{R}^{N_x N_y N_z \times p}.$$

It can be derived that

$$\mathbf{R}_m = -\mathbf{B}^{\mathrm{T}}\mathbf{B}\mathbf{C}_m. \tag{10}$$

Consequently, (9) can be compactly represented in matrix form

$$\bar{\mathbf{C}}^{(m)} = \mathbf{C}^{(m)} - \mathbf{C}_m\mathbf{\Omega}^{(m)}. \tag{11}$$

To determine the combination coefficient vector $\mathbf{\Omega}^{(m)}$ in (11), we minimize the $l^2$-norm of the residual of $\bar{\mathbf{C}}^{(m)}$

$$\begin{aligned}
\mathbf{\Omega}^{(m)} &= \arg\min_{\mathbf{\Omega}^{(m)} \in \mathbb{R}^p} \left\|\bar{\mathbf{r}}^{(m)}\right\|_2 \\
&= \arg\min_{\mathbf{\Omega}^{(m)} \in \mathbb{R}^p} \left\|\mathbf{B}^{\mathrm{T}}(\mathbf{b} - \mathbf{B}\bar{\mathbf{C}}^{(m)})\right\|_2 \\
&= \arg\min_{\mathbf{\Omega}^{(m)} \in \mathbb{R}^p} \left\|\mathbf{B}^{\mathrm{T}}[\mathbf{b} - \mathbf{B}(\mathbf{C}^{(m)} - \mathbf{C}_m\mathbf{\Omega}^{(m)})]\right\|_2 \\
&= \arg\min_{\mathbf{\Omega}^{(m)} \in \mathbb{R}^p} \left\|\mathbf{r}^{(m)} - \mathbf{R}_m\mathbf{\Omega}^{(m)}\right\|_2.
\end{aligned}$$

By setting the derivative of $\|\mathbf{r}^{(m)} - \mathbf{R}_m\mathbf{\Omega}^{(m)}\|_2$ to zero, we obtain

$$\mathbf{R}_m^{\mathrm{T}}\mathbf{R}_m\mathbf{\Omega}^{(m)} = \mathbf{R}_m^{\mathrm{T}}\mathbf{r}^{(m)}.$$

Under the assumption that $\mathbf{R}_m^T \mathbf{R}_m$ is full column rank, the optimal combination coefficient vector $\mathbf{\Omega}^{(m)}$ is given by

$$\mathbf{\Omega}^{(m)} = (\mathbf{R}_m^{\mathrm{T}} \mathbf{R}_m)^{-1} \mathbf{R}_m^{\mathrm{T}} \mathbf{r}^{(m)}.$$

From (11), we have

$$\bar{\mathbf{C}}^{(m)} = \mathbf{C}^{(m)} - \mathbf{C}_m (\mathbf{R}_m^{\mathrm{T}} \mathbf{R}_m)^{-1} \mathbf{R}_m^{\mathrm{T}} \mathbf{r}^{(m)}. \tag{12}$$

The iteration step defined in (12) for the asynchronous I-PIA is known as the Anderson extrapolation step for $\mathbf{C}^{(m)}$.

### 3.3 Alternating Anderson extrapolation with asynchronous I-PIA

For the Anderson extrapolation step $\bar{\mathbf{C}}^{(m)}$, we proceed to iterate on $\bar{\mathbf{C}}^{(m)}$ using the asynchronous I-PIA method, yielding

$$\mathbf{C}^{(m+1)} = \mathbf{C}^{(m)} + v_m \bar{\mathbf{r}}^{(m)}, \tag{13}$$

where $\bar{\mathbf{r}}^{(m)} = \mathbf{B}^T (\mathbf{b} - \mathbf{B} \bar{\mathbf{C}}^{(m)})$ represents the residual of $\bar{\mathbf{C}}^{(m)}$, and $v_m > 0$ is a variable step size dependent on $m$. From (10), (12), and (13), we have

$$\mathbf{C}^{(m+1)} = \mathbf{C}^{(m)} + (v_m \mathbf{I} - (\mathbf{C}^{(m)} + v_m \mathbf{R}_m)(\mathbf{R}_m^{\mathrm{T}} \mathbf{R}_m)^{-1} \mathbf{R}_m^{\mathrm{T}}) \mathbf{r}^{(m)}. \tag{14}$$

After conducting $q$ iterations of asynchronous I-PIA, the Anderson extrapolation technique is applied to extrapolate the $p + 1$ iterative sequences generated. The process then recommences with the Anderson extrapolation step, followed by $q$ additional iterations of asynchronous I-PIA. This method, which we term the alternating iterative approach between Anderson extrapolation and asynchronous I-PIA, generates a sequence of control coefficients for a B-spline implicit function. The overall procedure of this alternating iterative method can be summarized as

$$\mathbf{C}^{(m+1)} = \mathbf{C}^{(m)} + \mathbf{D}_m \mathbf{r}^{(m)},$$

where

$$\mathbf{D}_m = \begin{cases} \mu_m \mathbf{I} & \text{if } \left(\dfrac{m}{p}\right) \notin \mathbb{N}, \\[2ex] v_m \mathbf{I} - (\mathbf{C}^{(m)} + v_m \mathbf{R}_m)(\mathbf{R}_m^{\mathrm{T}} \mathbf{R}_m)^{-1} \mathbf{R}_m^{\mathrm{T}} & \text{if } \left(\dfrac{m}{p}\right) \in \mathbb{N}. \end{cases}$$

The step size $v_m$ in (14) can be determined by minimizing the squared $l^2$-norm of the residual of $\mathbf{C}^{(m+1)}$, that is,

$$v_m = \arg \min_{v_m \in \mathbb{R}} \left\| \mathbf{B}^{\mathrm{T}} (\mathbf{b} - \mathbf{B} \mathbf{C}^{(m+1)}) \right\|_2^2. \tag{15}$$

By substituting (14) into (15) and simplifying the resulting expressions, we have

$$v_m = \arg \min_{v_m \in \mathbb{R}} \left\| (\mathbf{I} - v_m \mathbf{B}^{\mathrm{T}} \mathbf{B})(\mathbf{I} - \mathbf{R}_m (\mathbf{R}_m^{\mathrm{T}} \mathbf{R}_m)^{-1} \mathbf{R}_m^{\mathrm{T}}) \mathbf{r}^{(m)} \right\|_2.$$

Let $\mathbf{N} = \mathbf{I} - \mathbf{R}_m(\mathbf{R}_m^{\mathrm{T}}\mathbf{R}_m)^{-1}\mathbf{R}_m^{\mathrm{T}}$ and define $f(v) = \left\|(\mathbf{I} - v\mathbf{B}^{\mathrm{T}}\mathbf{B})\mathbf{N}\mathbf{r}^{(m)}\right\|_2^2$. Then, $f(v)$ can be expanded as

$$
\begin{aligned}
f(v) \quad &= \left\|(\mathbf{I} - v\mathbf{B}^{\mathrm{T}}\mathbf{B})\mathbf{N}\mathbf{r}^{(m)}\right\|_2^2 \\
&= ((\mathbf{I} - v\mathbf{B}^{\mathrm{T}}\mathbf{B})\mathbf{N}\mathbf{r}^{(m)})^{\mathrm{T}}((\mathbf{I} - v\mathbf{B}^{\mathrm{T}}\mathbf{B})\mathbf{N}\mathbf{r}^{(m)}) \\
&= \left\|\mathbf{N}\mathbf{r}^{(m)}\right\|_2^2 - 2v(\mathbf{N}\mathbf{r}^{(m)})^{\mathrm{T}}(\mathbf{B}^{\mathrm{T}}\mathbf{B}\mathbf{N}\mathbf{r}^{(m)}) + v^2\left\|\mathbf{B}^{\mathrm{T}}\mathbf{B}\mathbf{N}\mathbf{r}^{(m)}\right\|_2^2.
\end{aligned}
$$

Setting the derivative of $f(v)$ with respect to $v$ to zero, we obtain the optimal $v_m$ given by

$$
v_m = \frac{(\mathbf{N}\mathbf{r}^{(m)})^{\mathrm{T}}(\mathbf{B}^{\mathrm{T}}\mathbf{B}\mathbf{N}\mathbf{r}^{(m)})}{\|\mathbf{B}^{\mathrm{T}}\mathbf{B}\mathbf{N}\mathbf{r}^{(m)}\|_2^2}. \tag{16}
$$

**Remark 3** *Building on the foundation established in [17], we in this paper introduce an innovative method for surface reconstruction that significantly enhances computational efficiency. Unlike the method described in [17], our proposed algorithm employs a dynamic step size strategy, where iteration step sizes are optimally adjusted in real time as the process evolves. This adaptive mechanism is essential, as it allows the algorithm to adjust its step sizes by minimizing the current residual, thus improving convergence, as discussed in [20]. This adaptive strategy not only enhances the convergence speed of the algorithm but also makes it more robust and suitable for a variety of problem characteristics. Consequently, we refer to this reconstruction algorithm with the dynamic step size strategy as the adaptive AA-I-PIA.*

### 3.4 Convergence analysis of the adaptive AA-I-PIA method

Next, we will discuss the convergence of the adaptive AA-I-PIA method.

**Lemma 1** *The I-PIA method converges when the step size $\mu$ satisfies*

$$
0 < \mu < \frac{2}{\lambda_{\max}(\mathbf{B}^{\mathrm{T}}\mathbf{B})}.
$$

**Theorem 1** *The adaptive AA-I-PIA method is convergent.*

**Proof 1** *When (16) holds, the iterative step $\mathbf{C}^{(m)}$ generated by the asynchronous I-PIA method is obtained based on minimizing the squared $l^2$-norm of the residual (denoted by $\mathbf{r}^{(m)}$), thus resulting in*

$$
\left\|\mathbf{r}^{(m)}\right\|_2 = \min_{\mu\in\mathbb{R}} \left\|(\mathbf{I} - \mu\mathbf{B}^{\mathrm{T}}\mathbf{B})\mathbf{r}^{(m-1)}\right\|_2.
$$

*It is evident that $\|\mathbf{r}^{(m)}\|_2$ is less than or equal to the squared $l^2$-norm of the residual $\tilde{\mathbf{r}}^{(m)}$ under the fixed step size strategy outlined in Lemma 1, that is,*

$$
\left\|\mathbf{r}^{(m)}\right\|_2 \le \left\|\tilde{\mathbf{r}}^{(m)}\right\|_2. \tag{17}
$$

*On one hand, from Eq (11), if $\omega_j^{(m)} = 0$ for $j = 1, 2, \cdots, p$, the Anderson extrapolation method of the asynchronous I-PIA simplifies to the asynchronous I-PIA method, yielding $\bar{\mathbf{C}}^{(m)} = \mathbf{C}^{(m)}$. On the other hand, since the Anderson extrapolation method minimizes the residual norm, we*

*have*

$$\left\| \bar{\mathbf{r}}^{(m)} \right\|_2 \leq \left\| \mathbf{r}^{(m)} \right\|_2. \tag{18}$$

*Combining* Eqs (17) and (18), *we conclude that*

$$\left\| \bar{\mathbf{r}}^{(m)} \right\|_2 \leq \left\| \mathbf{r}^{(m)} \right\|_2 \leq \left\| \tilde{\mathbf{r}}^{(m)} \right\|_2.$$

*By Lemma 1, the fixed step size I-PIA method converges, implying* $\lim_{m \to \infty} \|\tilde{\mathbf{r}}^{(m)}\|_2 = 0$, *and thus* $\lim_{m \to \infty} \|\bar{\mathbf{r}}^{(m)}\| = 0$. *Therefore, the adaptive AA-I-PIA method is convergent.*

## 4 Regularized adaptive implicit surface reconstruction algorithm

3D scanning devices are used to capture three-dimensional geometric information of objects or scenes and generate corresponding point cloud data. However, point cloud data often contains noise due to various factors such as sensor noise, environmental interference (strong light, shadows, reflections), motion artifacts, and uneven sampling density and distribution. Reconstructing surfaces directly from noisy data requires robust reconstruction algorithms that can handle noisy point cloud data. It is also important to effectively distinguish between noise points and valid geometric information while minimizing the impact of noise on the reconstruction results.

To mitigate the influence of noise on surface reconstruction algorithms, researchers typically employ preprocessing and noise filtering techniques to enhance the robustness of the reconstruction algorithms. Preprocessing refers to applying certain operations to the point cloud data before surface reconstruction to reduce the influence of noise. Common preprocessing methods include filtering techniques to remove noise or outliers and adjusting sampling density to achieve a more uniform and consistent point cloud data distribution. Noise filtering involves using dedicated noise filtering algorithms to effectively reduce the impact of noise on surface reconstruction. Noise filtering algorithms are designed based on the characteristics and distribution of noise to identify and remove noise points, thereby improving the accuracy of surface reconstruction.

In [17], it is mentioned that introducing regularization constraints in surface reconstruction algorithms can enhance noise suppression capabilities. Regularization constraints can be achieved by adding smoothness constraints, curvature constraints, or boundary constraints to the objective function. These constraints help make the surface reconstruction results smoother and more continuous, reducing the influence of noise. Additionally, the authors in [13, 15] propose that regularization methods can reduce the instability caused by data perturbations in reconstruction algorithms. Common regularization methods include Tikhonov regularization, mixed gradient Tikhonov regularization, sparse regularization, regularization based on singular value decomposition, and other improved methods.

### 4.1 Regularized asynchronous I-PIA

The incorporation of regularization techniques into implicit surface reconstruction algorithms, as mentioned in [17], significantly bolsters their noise handling capabilities. In this subsection, we delve into the regularized asynchronous I-PIA method to enhance the computational efficiency of the methodologies proposed in [17].

In contrast to the update strategy in (6), in the regularized I-PIA process, the control coefficients are updated using the following formula

$$C_{ijk}^{(m+1)} = (1 - \tau\hat{\mu}_m)C_{ijk}^{(m)} + \hat{\mu}_m r_{ijk}^{(m)}, \tag{19}$$

where $\tau > 0$ is the regularization parameter and $\hat{\mu}_m$ is a real number that varies with $m$.

The matrix form of (19) is denoted as

$$\begin{aligned}
\mathbf{C}^{(m+1)} &= (1 - \tau\hat{\mu}_m)\mathbf{C}^{(m)} + \hat{\mu}_m\mathbf{r}^{(m)} \\
&= [\mathbf{I} - \hat{\mu}_m(\tau\mathbf{I} + \mathbf{B}^{\mathrm{T}}\mathbf{B})]\mathbf{C}^{(m)} + \hat{\mu}_m\mathbf{B}^{\mathrm{T}}\mathbf{b}.
\end{aligned} \tag{20}$$

We refer to (20) as the regularized asynchronous I-PIA method. Similar to the step size selection in the asynchronous I-PIA method, the step size in the asynchronous regularized I-PIA method can also be selected by minimizing the squared $l^2$-norm of the residual, resulting in

$$\hat{\mu}_m = \frac{(\mathbf{r}^{(m)})^{\mathrm{T}}((\tau\mathbf{I} + \mathbf{B}^{\mathrm{T}}\mathbf{B})\mathbf{r}^{(m)})}{\|(\tau\mathbf{I} + \mathbf{B}^{\mathrm{T}}\mathbf{B})\mathbf{r}^{(m)}\|_2^2}.$$

## 4.2 Regularized adaptive AA-I-PIA

Similar to the method detailed in Section 3.3, we initially employ the regularized asynchronous I-PIA to generate several iterations of the sequence. We then apply the Anderson extrapolation method to produce extrapolated steps, which are used to continue the iterations with the regularized asynchronous I-PIA. By repeating this process, we obtain a sequence of control coefficients suitable for generating B-spline implicit functions. This alternating iterative approach is termed the regularized adaptive AA-I-PIA. As its derivation process closely mirrors that of the adaptive AA-I-PIA method outlined in Section 3.3, we omit the derivation here. The regularized adaptive AA-I-PIA is succinctly described by

$$\mathbf{C}^{(m+1)} = \begin{cases} (1 - \tau\hat{\mu}_m)\mathbf{C}^{(m)} + \hat{\mu}_m\mathbf{r}^{(m)} & \text{if } \left(\dfrac{m}{p}\right) \notin \mathbb{N}, \\[2ex] (1 - \tau\hat{v}_m)\mathbf{C}^{(m)} + \bar{\mathbf{D}}_{(m)}\mathbf{r}^{(m)} & \text{if } \left(\dfrac{m}{p}\right) \in \mathbb{N}, \end{cases}$$

where $\bar{\mathbf{D}}_{(m)} = v_m\mathbf{I} - [(1 - \tau\hat{v}_m)\mathbf{C}^{(m)} + \hat{v}_m\mathbf{R}_m](\mathbf{R}_m^{\mathrm{T}}\mathbf{R}_m)^{-1}\mathbf{R}_m^{\mathrm{T}}$, and $\mathbf{r}^{(m)} = \mathbf{B}^{\mathrm{T}}(\mathbf{b} - \mathbf{B}\mathbf{C}^{(m)})$.

It is noteworthy that the regularized adaptive AA-I-PIA method is derived by integrating the Anderson acceleration strategy with the regularized asynchronous I-PIA. Following a convergence analysis analogous to that of the adaptive AA-I-PIA, it can be inferred that if the regularized asynchronous I-PIA method converges, then the regularized adaptive AA-I-PIA method will also converge.

## 5 Numerical examples

In this section, we present the results of numerical experiments designed to validate the effectiveness of our proposed surface reconstruction algorithms. These experiments were conducted using Matlab, on a personal computer equipped with an Intel Core i7-13700F 2.10 GHz processor and 32 GB of RAM.

In the experiments, the point dataset $\mathbf{p}_r = (x_r, y_r, z_r)$, for $r = 1, 2, \cdots, n$ was normalized to a unit cube. The parameter $\sigma$ in (2) was set to 0.01 when generating additional offset points.

To ensure computational efficiency, the stopping criterion for the proposed iterative methods employed in our experiments is defined by the condition

$$\text{ERROR}^{(m)} < 1 \times 10^{-6} \quad \text{or} \quad \text{ERROR}^{(m)} \geq \frac{\theta}{p+1} \sum_{i=m-p-1}^{m-1} \text{ERROR}^{(m-1)}, \ \theta \in (0, 1),$$

where $\text{ERROR}^{(m)}$ represents the fitting error of the reconstructed surface after $m$ iterations, calculated as the infinity norm of the residual vector $[\delta_1, \delta_2, \cdots, \delta_{3n}]^T$. The parameter $\theta$ is set to 0.999 for these experiments. If other methods' error levels match or exceed that of the proposed method or if they exceed 10,000 iterations, the process is terminated.

In constructing the implicit B-spline function, we define a cubic grid with dimensions $N_x \times N_y \times N_z$, where $N_x$, $N_y$, and $N_z$ correspond to the number of basis functions. In the experiments, we utilize an equal number of basis functions for all dimensions, i.e., $N_x = N_y = N_z$.

## 5.1 Surface reconstruction results of accurate point cloud data

In this section, we evaluate the computational efficiency of our adaptive step size surface reconstruction algorithm by applying it to four high-precision point cloud data models: Kitten, Skeleton, Max Planck, and Armadillo. The performance of our method is compared against the I-PIA algorithm [12] and the AA-I-PIA algorithm [17]. For these models, the B-spline grid sizes were chosen as $40 \times 40 \times 40$, $60 \times 60 \times 60$, $100 \times 100 \times 100$, and $150 \times 150 \times 150$, respectively. Table 1 presents a comparison of the number of iterations and runtime required by the three methods to achieve similar levels of reconstruction error. Fig 1 depicts the fitting errors versus the number of iterations. The results demonstrate that the adaptive step size strategy employed in our AA-I-PIA algorithm leads to a reduction in both the number of iterations and computational time, outperforming both the fixed step size strategy used in AA-I-PIA and the I-PIA algorithm in terms of convergence rate and computational efficiency.

Figs 2 to 5 illustrate the reconstructed surfaces of the Kitten, Skeleton, Max Planck, and Armadillo models using both the I-PIA and the adaptive AA-I-PIA methods. Both methods effectively fit the point cloud data. The results consistently show that the adaptive AA-I-PIA significantly enhances the convergence speed compared to the traditional I-PIA method.

## 5.2 Surface reconstruction results of point cloud data with missing data

During point cloud data acquisition, partial data loss is common due to factors such as sensor noise, occlusions, or sensor damage. The robustness of I-PIA and AA-I-PIA in handling such losses has been noted by authors in [12, 15]. These methods are known for effectively managing and compensating for data gaps. The following examples demonstrate the efficiency of an adaptive algorithm designed for surface reconstruction. This algorithm is tested on point cloud data with missing values to validate its ability to handle real-world scenarios with

**Table 1. Comparison of adaptive AA-I-PIA with I-PIA [12] and AA-I-PIA [17].**

| Models | Points count | Grids count | ERROR$^{(m)}$ | Adaptive AA-I-PIA | | AA-I-PIA | | I-PIA | |
|---|---|---|---|---|---|---|---|---|---|
| | | | | IT | CPU | IT | CPU | IT | CPU |
| Kitten | 11039 | $40 \times 40 \times 40$ | $7.59 \times 10^{-3}$ | 390 | 1.97 | 2266 | 8.17 | 37572 | 109.51 |
| Skeleton | 32644 | $60 \times 60 \times 60$ | $7.23 \times 10^{-3}$ | 377 | 13.89 | 1966 | 41.81 | 32560 | 587.93 |
| Max Planck | 50112 | $100 \times 100 \times 100$ | $2.38 \times 10^{-3}$ | 285 | 21.00 | 1231 | 53.96 | 20315 | 671.22 |
| Armadillo | 65704 | $150 \times 150 \times 150$ | $2.95 \times 10^{-3}$ | 419 | 79.63 | 931 | 90.09 | 14769 | 859.60 |

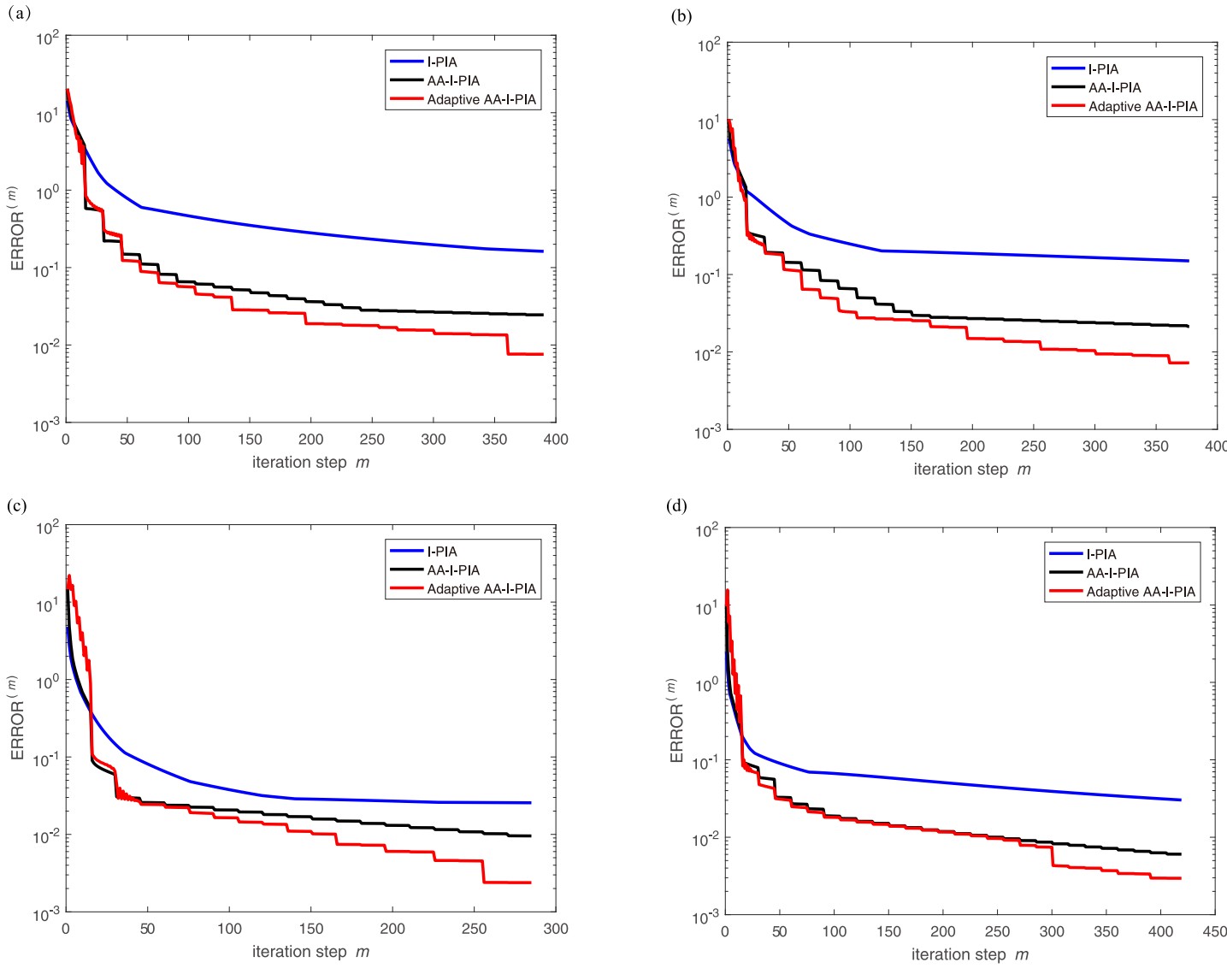

**Fig 1. Comparison of fitting error vs number of iterations.**

compromised data integrity. These demonstrations aim to highlight the potential of the algorithms to improve the reliability and precision of point cloud reconstruction, even amidst data loss.

The surfaces of the Girl model, which features a single crack, and the BU model, with three local data loss instances (referred to as "holes"), were meticulously reconstructed using adaptive AA-I-I-PIA, I-PIA, and AA-I-PIA methods. The results are detailed in Table 2. For the Girl model, the adaptive AA-I-PIA required only 448 iterations and completed the process in 28.98 seconds, significantly outperforming the AA-I-PIA, which needed 1546 iterations and 50.89 seconds, and the I-PIA, requiring 26,267 iterations and 1087.39 seconds. Similarly, for the BU model with holes, the adaptive AA-I-PIA demonstrated superior efficiency with 795 iterations and 68.32 seconds, compared to AA-I-PIA's 1696 iterations and 95.16 seconds, and I-PIA's extensive 29,045 iterations and 1322.42 seconds. These results underscore the adaptive

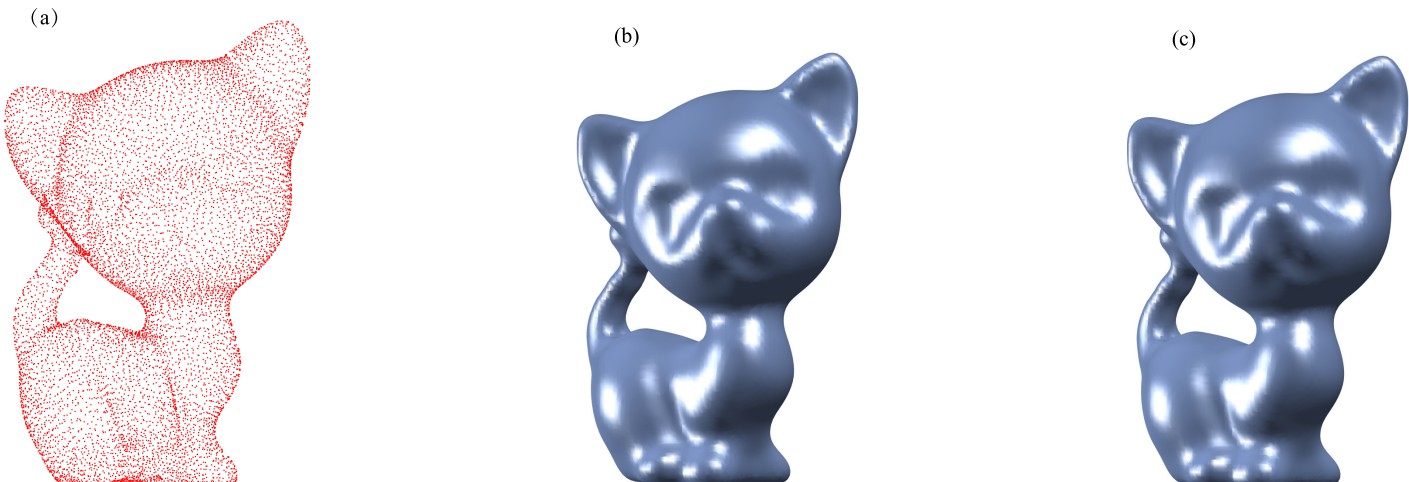

**Fig 2. Surface reconstruction of the Kitten model, which includes 11,039 data points.** (a) illustrates the initial point cloud sampled from the Kitten model. (b) and (c) display the reconstructed surfaces after 37,572 iterations of the I-PIA method and 390 iterations of the adaptive AA-I-PIA method, respectively.

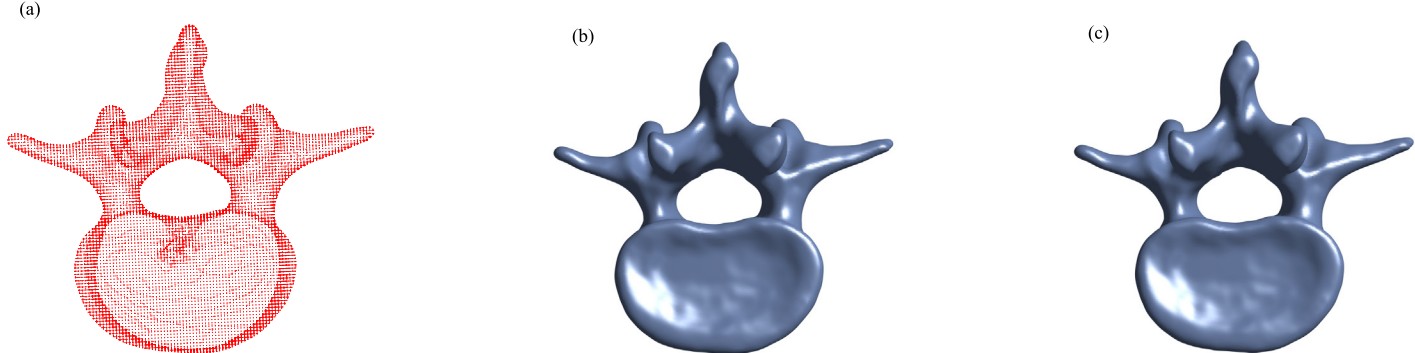

**Fig 3. Surface reconstruction of the Skeleton model, which includes 32,644 data points.** (a) displays the initial point cloud sampled from the Skeleton model. (b) and (c) present the reconstructed surfaces after 32,560 iterations of the I-PIA method and 377 iterations of the adaptive AA-I-PIA method, respectively.

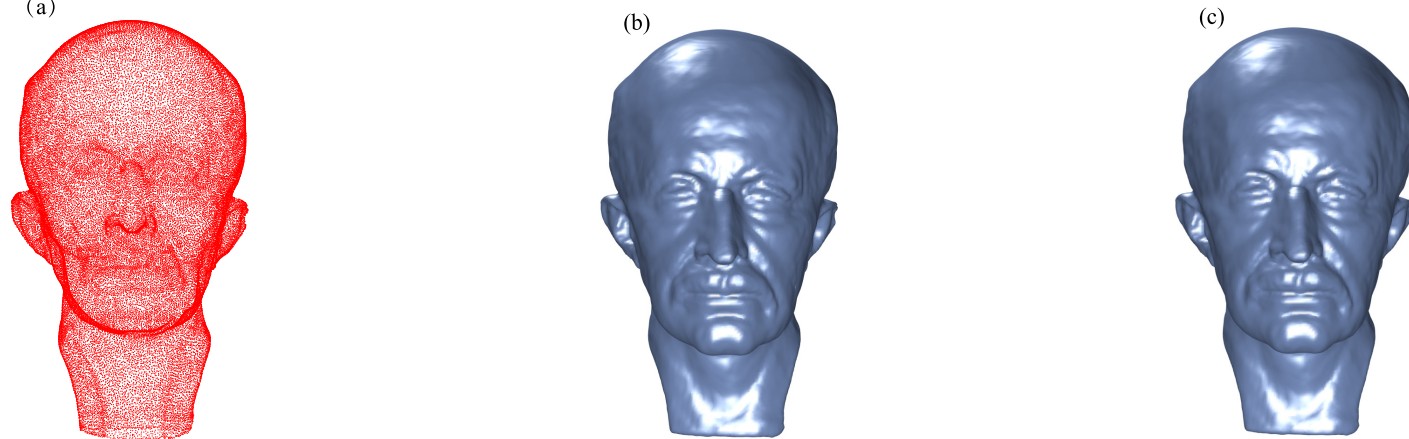

**Fig 4. Surface reconstruction of the Max Planck model, which includes 50,112 data points.** (a) presents the initial point cloud sampled from the Max Planck model. (b) and (c) showcase the reconstructed surfaces after 20,315 iterations of the I-PIA method and 285 iterations of the adaptive AA-I-PIA method, respectively.

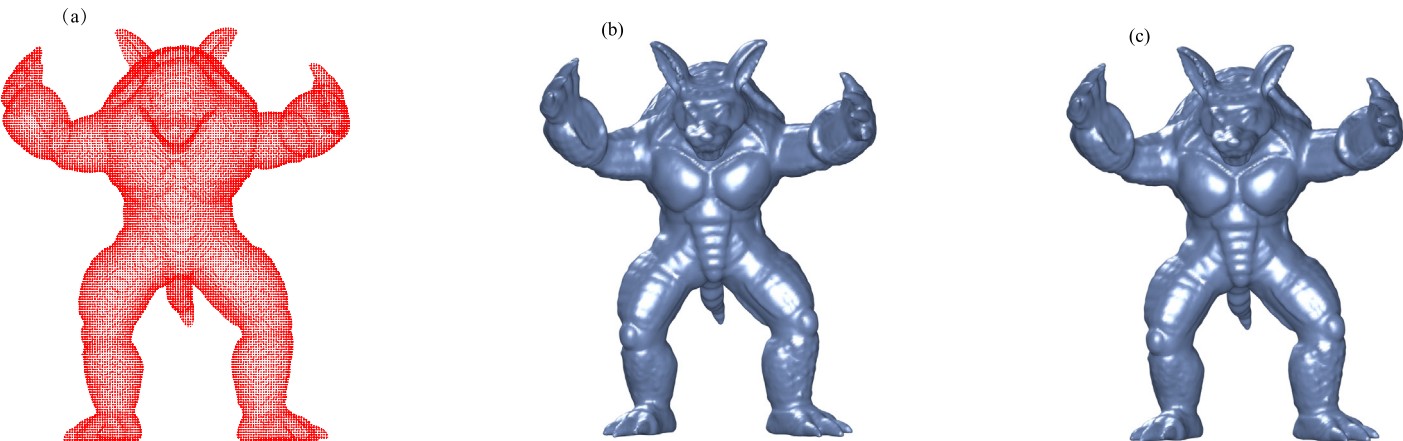

**Fig 5. Surface reconstruction of the Armadillo model, which includes 65,704 data points.** (a) displays the initial point cloud sampled from the Armadillo model. (b) and (c) illustrate the reconstructed surfaces after 14,769 iterations of the I-PIA method and 419 iterations of the adaptive AA-I-PIA method, respectively.

**Table 2. Comparison of reconstruction algorithms for point cloud data with missing data.**

| Models | Points count | Grids count | ERROR$^{(m)}$ | Adaptive AA-I-PIA | | AA-I-PIA | | I-PIA | |
|---|---|---|---|---|---|---|---|---|---|
| | | | | IT | CPU | IT | CPU | IT | CPU |
| Girl | 72769 | $60 \times 60 \times 60$ | $1.02 \times 10^{-2}$ | 448 | 28.98 | 1546 | 50.89 | 26267 | 1087.39 |
| BU | 73382 | $100 \times 100 \times 100$ | $2.88 \times 10^{-3}$ | 795 | 68.32 | 1696 | 95.16 | 29045 | 1322.42 |

AA-I-PIA's superiority in handling point cloud models with missing data, offering a highly efficient solution for surface reconstruction tasks.

The point cloud models and their reconstruction outcomes are displayed in Figs 6 and 7. It is evident that both methods effectively conform to the point cloud data and successfully fill in the missing regions.

In summary, the adaptive AA-I-PIA surpasses I-PIA and AA-I-PIA in handling missing data by offering faster convergence, and greater computational efficiency. These advantages

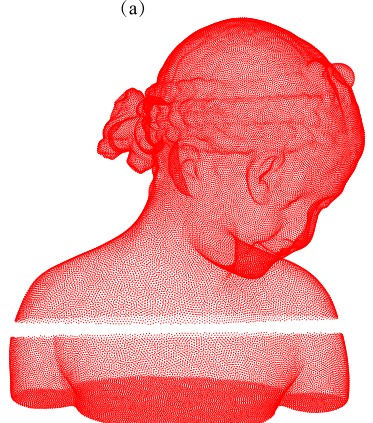
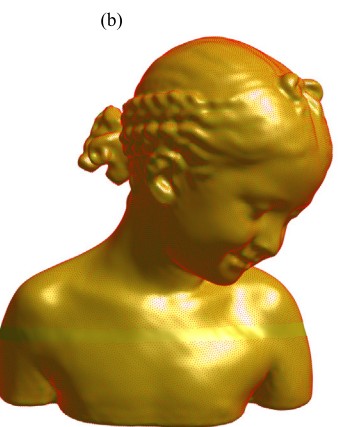
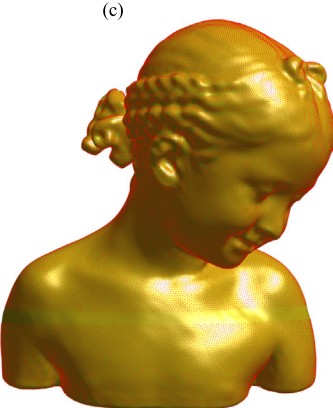

**Fig 6. Surface reconstruction of the Girl model with a crack, consisting of 72,769 data points.** (a) shows the initial points sampled from the Girl model. (b) and (c) depict the reconstructed surfaces obtained after 26,267 iterations of I-PIA and 448 iterations of the regularized adaptive AA-I-PIA, respectively.

(a) 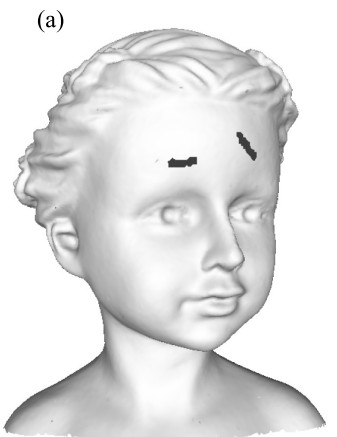
(b) 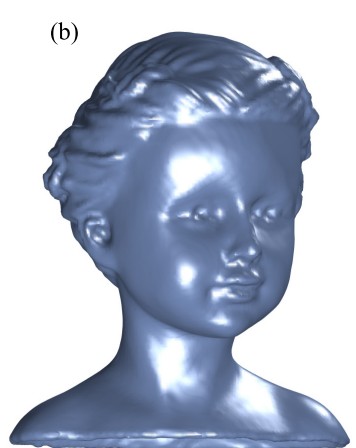
(c) 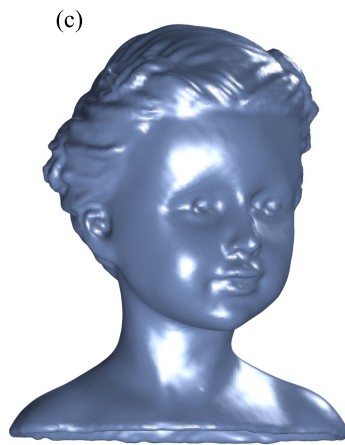

**Fig 7. Surface reconstruction of the BU model with two holes, consisting of 73,382 data points.** (a) shows the initial points sampled from the BU model. (b) and (c) depict the reconstructed surfaces obtained after 29,045 iterations of I-PIA and 795 iterations of the regularized adaptive AA-I-PIA, respectively.

position the adaptive AA-I-PIA as the optimal choice for applications that demand swift and accurate reconstruction from incomplete point cloud data.

## 5.3 Surface reconstruction results of point cloud data with noise

Three experiments involving noisy data were conducted using the Duck, Bunny, and Skeleton point cloud models, which contain 9,640, 10,680, and 32,644 data points, respectively. Perturbations of 0.001, 0.002, and 0.002 were introduced to these datasets using MATLAB's built-in 'rand' function. We then reconstructed the noisy data using both the adaptive and the regularized adaptive AA-I-PIA, comparing their performance with that of the regularized AA-I-PIA and I-PIA. The B-spline control coefficient grids were set to $30 \times 30 \times 30$, $50 \times 50 \times 50$, and $60 \times 60 \times 60$ for the Duck, Bunny, and Skeleton models, respectively. These experiments aimed to assess the algorithms' adaptability to noise and evaluate their reconstruction performance.

Table 3 presents a detailed comparison of the numerical results for the adaptive AA-I-PIA, AA-I-PIA, I-PIA, and their regularized counterparts. In this table, "–" signifies that the iteration count exceeded 500,000, leading to termination. It is evident that regularized reconstruction methods outperform non-regularized ones in terms of iteration count and computation time. Moreover, the proposed adaptive method significantly outperforms the methods described in [7]. Fig 8 displays the noisy point cloud data and the surfaces reconstructed using the regularized adaptive AA-I-PIA method, demonstrating the high computational efficiency of the fast algorithm with an adaptive step size in handling noisy data and providing an effective solution for reconstruction challenges.

**Table 3. Comparison of reconstruction results for point cloud model with noise.**

| Models | ERROR$^{(m)}$ | Regularized adaptive AA-I-PIA | | Adaptive AA-I-PIA | | Regularized AA-I-PIA | | Non-regularized AA-I-PIA | | Regularized I-PIA | | Non-regularized I-PIA | |
|---|---|---|---|---|---|---|---|---|---|---|---|---|---|
| | | IT | CPU | IT | CPU | IT | CPU | IT | CPU | IT | CPU | IT | CPU |
| Duck | $6.00 \times 10^{-4}$ | 1425 | 1.23 | 323116 | 316.28 | 6931 | 3.71 | – | – | 117860 | 229.31 | – | – |
| Bunny | $2.29 \times 10^{-7}$ | 1621 | 13.83 | – | – | 12526 | 68.44 | – | – | 207849 | 2041.89 | – | – |
| Skeleton | $1.38 \times 10^{-2}$ | 166 | 4.59 | 394 | 12.02 | 361 | 8.65 | 1099 | 21.84 | 5401 | 177.77 | 18032 | 586.29 |

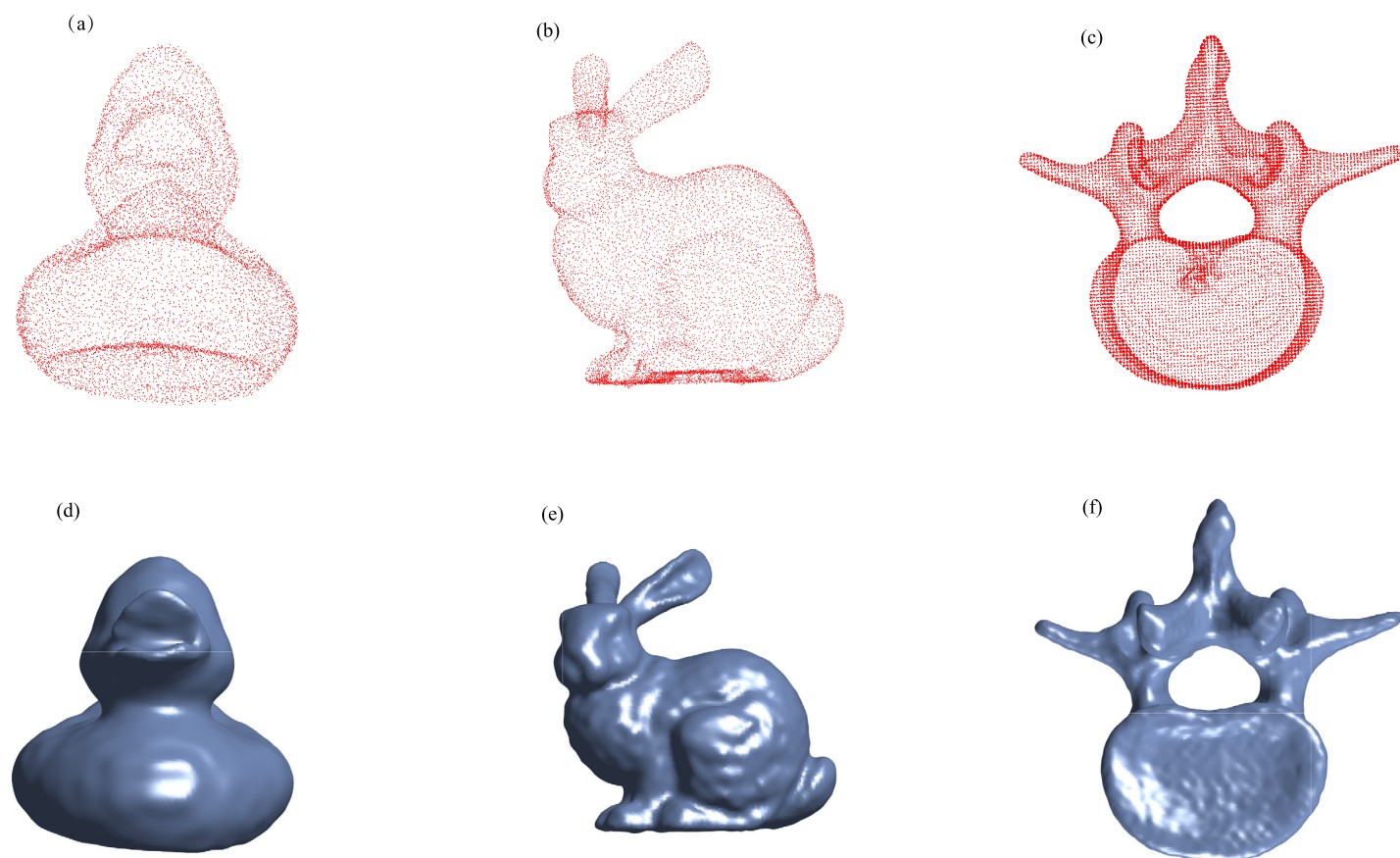

**Fig 8. Noisy point cloud models and reconstructed surfaces generated by the regularized adaptive AA-I-PIA.**

## 6 Conclusion

This paper presents an extensive investigation into the surface reconstruction algorithm initially proposed in [17], with particular emphasis on the innovative adaptive method for step size selection. Addressing the challenge of noise in point cloud data, the paper further introduces an adaptive regularized algorithm for fast surface reconstruction. The adaptive method has been shown to excel in tackling the surface reconstruction challenge associated with point cloud data that includes random noise. A significant reduction in the number of iterations and the overall computation time has been achieved without compromising the reconstruction error, thereby effectively enhancing the computational efficiency of the surface reconstruction process. The experimental results also demonstrate the superiority of the adaptive step size selection strategy over the fixed step size method previously described in [17]. This advancement is particularly beneficial for implicit B-spline surface reconstruction, offering a more efficient solution that maintains the integrity and accuracy of the reconstructed surfaces.

In conclusion, the paper's contributions are twofold: the innovative adaptive step size selection and the development of an adaptive regularized algorithm. These innovations have been substantiated through rigorous experimental validation. They not only overcome the shortcomings of existing methodologies but also set the stage for more potent and efficient surface reconstruction techniques, even in noisy environments. This work, therefore, contributes to the advancement of computational geometry and its manifold applications.

Future work will explore how to integrate preconditioning techniques into our surface reconstruction algorithms. We will investigate the effects of the preconditioning technique on convergence and computational efficiency. It is hoped that through these studies, we can further enhance the robustness of the algorithms, expand their application range in complex scenarios, and contribute new perspectives and solutions to the field of surface reconstruction.

## Acknowledgments

We would like to thank Professor Hongwei Lin from Zhejiang University and Dr. Haibo Wang from Central South University for providing the point cloud datasets. Their contributions have been invaluable to the success of this research and the validity of the results.

## Author Contributions

**Funding acquisition:** Jingguo Dai.

**Methodology:** Chengzhi Liu.

**Resources:** Yeqing Yi.

**Supervision:** Jingguo Dai, Chengzhi Liu.

**Writing – original draft:** Chengzhi Liu.

**Writing – review & editing:** Yeqing Yi.

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
