## [Decision Letter · Decision Letter 0]

9 Oct 2024

PONE-D-24-37888Fast Surface Reconstruction Algorithm with Adaptive Step SizePLOS ONE

Dear Dr. Liu,

Thank you for submitting your manuscript to PLOS ONE. After careful consideration, we feel that it has merit but does not fully meet PLOS ONE’s publication criteria as it currently stands. Therefore, we invite you to submit a revised version of the manuscript that addresses the points raised during the review process. Please provide response to comments point by point. Please submit your revised manuscript by Nov 23 2024 11:59PM. If you will need more time than this to complete your revisions, please reply to this message or contact the journal office at plosone@plos.org. Please include the following items when submitting your revised manuscript:A rebuttal letter that responds to each point raised by the academic editor and reviewer(s). You should upload this letter as a separate file labeled 'Response to Reviewers'.A marked-up copy of your manuscript that highlights changes made to the original version. You should upload this as a separate file labeled 'Revised Manuscript with Track Changes'.An unmarked version of your revised paper without tracked changes. You should upload this as a separate file labeled 'Manuscript'.If applicable, we recommend that you deposit your laboratory protocols in protocols.io to enhance the reproducibility of your results. Protocols.io assigns your protocol its own identifier (DOI) so that it can be cited independently in the future. For instructions see: https://journals.plos.org/plosone/s/submission-guidelines#loc-laboratory-protocols. Additionally, PLOS ONE offers an option for publishing peer-reviewed Lab Protocol articles, which describe protocols hosted on protocols.io. Read more information on sharing protocols at https://plos.org/protocols?utm_medium=editorial-email&utm_source=authorletters&utm_campaign=protocols.

We look forward to receiving your revised manuscript.

Kind regards,

Xuebo Zhang, Ph.D.

Academic Editor

PLOS ONE

Journal Requirements:

"This work was supported in part by the Key-Area Research and Development Program of Guangdong Province

under Grant 2022B0101020002, in part by the Key Area Special Programs for Guangdong Province Universities

under Grant 2022ZDZX4043, Shaoguan City Science and Technology Plan Projects under Grant 220606154533881,

and in part by the Scientific Research Funds of Hunan Provincial Education Department under Grant 21B0790."

7. We note that your Data Availability Statement is currently as follows: All relevant data are within the manuscript and its Supporting Information files.

8. We note that Figures 1, 2, 3, 4, 5, 6 and 7 in your submission contain copyrighted images. All PLOS content is published under the Creative Commons Attribution License (CC BY 4.0), which means that the manuscript, images, and Supporting Information files will be freely available online, and any third party is permitted to access, download, copy, distribute, and use these materials in any way, even commercially, with proper attribution. For more information, see our copyright guidelines: http://journals.plos.org/plosone/s/licenses-and-copyright.

a. You may seek permission from the original copyright holder of Figures 1, 2, 3, 4, 5, 6 and 7 to publish the content specifically under the CC BY 4.0 license. 

Reviewers' comments:

Reviewer's Responses to Questions

**Comments to the Author**

1. Is the manuscript technically sound, and do the data support the conclusions?

Reviewer #1: Yes

Reviewer #2: Yes

Reviewer #3: Yes

2. Has the statistical analysis been performed appropriately and rigorously? 

Reviewer #1: Yes

Reviewer #2: Yes

Reviewer #3: N/A

3. Have the authors made all data underlying the findings in their manuscript fully available?

Reviewer #1: No

Reviewer #2: Yes

Reviewer #3: Yes

4. Is the manuscript presented in an intelligible fashion and written in standard English?

Reviewer #1: Yes

Reviewer #2: No

Reviewer #3: Yes

5. Review Comments to the Author

Reviewer #1: Title:- Fast Surface Reconstruction Algorithm with Adaptive Step Size

Reviewer:- In this study, authors have proposed adaptive method for step size selection in surface reconstruction algorithm. This article is interesting but needs to be revised by considering the following points.

1. There is some inconsistency in section list (e.g As per the last part of introduction section, Numerical examples section must be given number 5.Similarly conclusion part is on number 6). Moreover, needs to be revised for correcting grammatically and typo mistakes.

2. Suggest to add preliminaries about adaptive step size in basic section.

3. This article is based on adaptive step size, so needs to be add some recent articles relevant to adaptive step size in introduction section.

4. It would be better if you define comparison of various algorithms (Iterations & computational complexity) in the form of continuous graph.

5. - To facilitate reproducible research, if possible, I suggest that the authors release the related source codes on github.com, the website of the authors' research group, or a similar website. This could make a positive impact on the academic community.

6. Needs to be add future motivation in the last paragraph of conclusion.

Reviewer #2: In this paper, the authors propose and investigate a surface reconstruction algorithm with an adaptive step size to enhance the computational efficiency of their previous work on Anderson-accelerated implicit progressive iterative approximation (I-PIA). Specifically, the step size is adjusted based on the current residual: larger residuals suggest the need for larger step sizes, while smaller residuals permit smaller step sizes.

The findings presented are both new and intriguing, and I am confident in the rigor of the analysis and numerical experiments. However, a notable weakness of this paper is that it does not offer substantial new insights. If the "adaptive step size" innovation is deemed sufficient for PLOS ONE, I will endorse the acceptance of this paper for publication, contingent upon a thorough review of its formatting, including things like line indents and punctuation.

Reviewer #3: Fast Surface Reconstruction Algorithm with Adaptive Step Size

Dr. Chengzhi Liu

Summary

In this paper a surface reconstruction algorithm is proposed. It is an accelerated version of the I-PIA method of the author as given in Reference 1. The new part is an adaptation of the stepsize of the used Anderson Acceleration method in order to speed up the convergence. From various numerical experiments it follows that the new algorithm is faster than the original method.

Comment

After minor revision this manuscript is suitable for publication in PLOS ONE. Motivation for this recommendation is given below.

Remarks:

1. I miss a couple of references to the literature, please add these in the revised manuscript:

- please give a reference to the paper where the Anderson Acceleration method is proposed first:

D.G. Anderson, Iterative procedures for nonlinear integral equations, J. ACM 12 (1965) 547-560

- please give references to papers where variable Anderson is used:

D.G.M. Anderson, Comments on "Anderson acceleration, mixing and extrapolation", Numer. Algorithms 80 (1) (2019) 135-234

F.A. Potra, H. Engler, A characterization of the behavior of the Anderson acceleration on linear problems, linear Algebra Appl. 438 (3) (2013) 1002-1011

K. Chen and C. Vuik, Non-stationary Anderson acceleration with optimized damping Journal of Computational and Applied Mathematics, 451, 116077, 2024

2. 4 lines below equation (6): is the matrix B^TB non-singular? If yes please give a prove, if no how should singular systems be handled?

3. 3 lines below equation (9) the relation with an iterative methods is mentioned. I think that the Richardon iteration method is ment. If yes please give a reference, otherwise indicate which method is ment.

4. One line below equation (15): what happens if matrix Omega is not of full rank?

5. Typo: section Numerical Examples should be Section 5.

6. Which programming language is used for the experiments?

6. PLOS authors have the option to publish the peer review history of their article (what does this mean?). If published, this will include your full peer review and any attached files.

Reviewer #1: **Yes: **Khurram shahzad, Shanghai university China. Email:- raokamran770@gmail.com

Reviewer #2: **Yes: **Kewang Chen

Reviewer #3: No

---

## [Author Response · Author response to Decision Letter 0]

7 Nov 2024

We would like to thank you for their worthy comments and suggestions on our manuscript.

We have revised our manuscript after reading the suggestions provided by the reviewers.

We listed below the detailed changes that we have made in the revised manuscript in response to the reviewers' suggestions.

We hope that the revised manuscript has addressed all of the reviewers' suggestions.

!

---

## [Decision Letter · Decision Letter 1]

18 Nov 2024

Fast Surface Reconstruction Algorithm with Adaptive Step Size

PONE-D-24-37888R1

Dear Dr. Liu,

We’re pleased to inform you that your manuscript has been judged scientifically suitable for publication and will be formally accepted for publication once it meets all outstanding technical requirements.

Kind regards,

Xuebo Zhang, Ph.D.

Academic Editor

PLOS ONE

Additional Editor Comments (optional):

Reviewers' comments:

Reviewer's Responses to Questions

**Comments to the Author**

1. If the authors have adequately addressed your comments raised in a previous round of review and you feel that this manuscript is now acceptable for publication, you may indicate that here to bypass the “Comments to the Author” section, enter your conflict of interest statement in the “Confidential to Editor” section, and submit your "Accept" recommendation.

Reviewer #1: All comments have been addressed

Reviewer #3: All comments have been addressed

2. Is the manuscript technically sound, and do the data support the conclusions?

Reviewer #1: Yes

Reviewer #3: Yes

3. Has the statistical analysis been performed appropriately and rigorously? 

Reviewer #1: Yes

Reviewer #3: N/A

4. Have the authors made all data underlying the findings in their manuscript fully available?

Reviewer #1: Yes

Reviewer #3: Yes

5. Is the manuscript presented in an intelligible fashion and written in standard English?

Reviewer #1: Yes

Reviewer #3: Yes

6. Review Comments to the Author

Reviewer #1: Suggest to Authors to publish code relevant to this manuscript on Github or any other scientific platform.

Reviewer #3: No additional remarks for the authors. All points I have raised are taken into account. In my opinion the paper can be publisded.

7. PLOS authors have the option to publish the peer review history of their article (what does this mean?). If published, this will include your full peer review and any attached files.

Reviewer #1: **Yes: **Khurram Shahzad, School of Communication and Information Engineering, Shanghai University, China.

Reviewer #3: **Yes: **Cornelis Vuik

---

## [Editor Report · Acceptance letter]

13 Jan 2025

PONE-D-24-37888R1 

PLOS ONE

Dear Dr. Liu, 

I'm pleased to inform you that your manuscript has been deemed suitable for publication in PLOS ONE. Congratulations! Your manuscript is now being handed over to our production team.

Kind regards, 

on behalf of

Professor Xuebo Zhang 

Academic Editor

PLOS ONE